

# Liver stiffness and serum galectin 3 level significantly increases in patients with prediabetes: a fibroscan study

Huseyin Ali Ozturk and Hilmi Erdem Sumbul

Internal Medicine, Department of Internal Medicine, University of Health Sciences—Adana Health Practice and Research Center, Adana, Turkey

## ABSTRACT

**Aim:** Diabetes mellitus (DM) is associated with the development and progression of metabolic dysfunction-associated steatotic liver disease (MASLD). In our study, we aimed to evaluate the relationship between liver stiffness (LS) and galectin-3 levels in patients diagnosed with prediabetes.

**Methods:** A total of 120 participants were included in this prospective and cross-sectional study, comprising 40 patients with prediabetes, 40 patients with type 2 DM and 40 individuals with normal glucose metabolism. Human galectin-3 levels were measured using Enzyme-Linked Immunosorbent Assay (ELISA) with Human Galectin-3 kits. LS measurements were performed using the FibroScan® Mini 430 device (Echosens, France).

**Results:** In our study, we found that Fib-4 index, LS and galectin-3 levels were increased in patients with prediabetes. Another significant finding was that hemoglobin A1c (HbA1c) and galectin-3 levels were higher in patients with LS value ≥8 kPa and both HbA1c and galectin-3 levels were independently associated with LS.

**Conclusion:** Considering the increased prevalence of MASLD in prediabetes, we recommend early assessment of LS and measurement of galectin-3 levels in these patients.

Corresponding author
Hilmi Erdem Sumbul,
erdemsumbul@gmail.com

## INTRODUCTION

Diabetes mellitus (DM) is a chronic metabolic disease characterized by high blood glucose levels. According to the International Diabetes Federation, it is estimated that approximately 700 million people will suffer from type 2 DM by 2045 (*Saeedi et al., 2019*). It can cause damage to the heart, eyes, kidneys, blood vessels and nerves. The liver, being an organ with an important role in glucose homeostasis, is also associated with DM. Metabolic dysfunction-related steatotic liver disease (MASLD) is the most common liver disease worldwide and affects 30% of the population (*Younossi et al., 2023*). It includes liver disease associated with metabolic and cardiovascular disorders such as obesity, insulin resistance, hypertension, dyslipidemia and DM. MASLD is considered as a hepatic manifestation of metabolic syndrome (*Godoy-Matos, Silva Júnior & Valerio, 2020*). DM is

significantly associated with the development and progression of MASLD (*European Association for the Study of the Liver (EASL), European Association for the Study of Diabetes (EASD) & European Association for the Study of Obesity (EASO), 2024*). DM is also strongly linked to liver fibrosis in chronic liver disease. It is a major risk factor for the progression of liver fibrosis in patients with biopsy-proven disease (*Koehler et al., 2016*).

The most definitive diagnostic method for detecting and grading organ damage in the liver is biopsy. However, it is an invasive procedure with risks such as bleeding and infection. Considering the prevalence of the disease in the population, FibroScan has been developed as a non-invasive method for detecting liver fibrosis. Studies have shown that FibroScan has high performance in diagnosing fibrosis in patients with MASLD (*Zoncapè, Liguori & Tsochatzis, 2024*).

Galectins are lectins that have a strong affinity for β-galactosidases and contain at least one carbohydrate recognition domain. Studies have indicated that they are involved in processes such as angiogenesis, cell-cell adhesion, cell-matrix interaction, cell division, cell proliferation, apoptosis, inflammation, fibrosis and tumor development (*Dumic, Dabelic & Flögel, 2006*). Interest in galectin-3, which plays a role in liver fibrosis, has increased exponentially in recent years. Research has also shown that galectin-3 levels are associated with the pathophysiology of diabetic patients (*Atalar et al., 2019*). However, the role of galectin-3 in diabetes remains unclear and further studies are needed.

Prediabetes is a condition where blood glucose levels are higher than normal but not high enough to be diagnosed as DM. Microvascular complications such as retinopathy, neuropathy and nephropathy can also be observed in individuals with prediabetes (*Koc & Sumbul, 2019*). The prevalence of MASLD is significantly higher in individuals with prediabetes compared to those without glucose intolerance (*Godoy-Matos et al., 2024*).

Studies have shown an increase in liver stiffness (LS) measured by liver elastography in patients with type 2 diabetes. However, studies related to LS in prediabetic patients are limited. Additionally, studies showing the relationship between LS and galectin-3 levels in patients diagnosed with prediabetes is very scarce. In our study, we aimed to evaluate the relationship between LS and galectin-3 levels in patients diagnosed with prediabetes.

## MATERIAL AND METHOD

### Study population and laboratory measurements

Our prospective and cross-sectional study comprised 120 participants: 40 individuals diagnosed with prediabetes, meeting inclusion criteria based on medical history, physical examination, and prior tests; 40 patients with type 2 DM exhibiting hemoglobin A1c (HbA1c) levels between 6.5% and 8.5%, a minimum diabetes duration of 5 years, and absence of diabetes-related complications; and 40 individuals constituting the healthy control group with normal glucose metabolism. Normal glucose metabolism (NGM) is characterised by fasting blood glucose (FBG) levels below 100 mg/dL and HbA1c values below 5.7%. Prediabetes is characterised by fasting blood glucose levels ranging from 100 to 125 mg/dL or HbA1c values between 5.7% and 6.4%. DM was characterised by FBG values of ≥126 mg/dL, HbA1c levels of ≥6.5% or the administration of anti-diabetic medication (*American Diabetes Association, 2010*).

Patients with chronic renal failure, acute and chronic liver diseases, thyroid disorders, rheumatic diseases, malignancies, peripheral vascular and cerebrovascular diseases, valvular heart disease, heart failure, active infections, pregnant women, and those who declined participation were excluded from the study. The research was performed in compliance with the Declaration of Helsinki and received approval from the institutional ethics committee. All participants received a comprehensive explanation of the written permission forms and engaged in the study after providing signed informed consent. A comprehensive medical history and physical assessment were conducted. The Ethics Committee of Adana City Training and Research Hospital sanctioned the study under decision number 3,032, dated December 21, 2023. Following a 5-min period of repose in a subdued and tranquil setting, blood pressure readings were obtained from both arms utilising an appropriate cuff and pulse rates were assessed. Anthropometric measures of body weight were conducted. The body mass index (BMI) is determined by dividing body weight in kilogrammes by the square of height in meters (BMI = kg/m$^2$). The laboratory procedures of the study were conducted in the Biochemistry Laboratory at Health Sciences University Adana Training and Research Hospital. The biochemistry laboratory has the International Organization for Standardization quality control management system and The International Federation of Clinical Chemistry and Laboratory Medicine standards. Venous blood was collected from the antecubital vein following a minimum of 8 h of overnight fasting from both the patients and the control group during regular assessments. Participants' laboratory measurements were conducted utilising automated laboratory procedures (Abbott Aeroset, Minneapolis, MN, USA) and suitable commercial kits (Abbott, Minneapolis, MN, USA). The FIB-4 score was computed *via* the formula:

The FIB $-$ 4 score is calculated using the formula: $(\text{Age} \times \text{AST})/[\text{Platelet count} \times (\text{ALT}) \wedge (1/2)]$.

Alongside the standard blood tests, both the study participants and the control group had an extra tube of gel with a yellow cap extracted to gather blood for the research. The samples were rapidly transported to the laboratory, where they were centrifuged at 4,000 rpm for 10 min to isolate the serum components. Prior to measurement, the samples were stored in an Eppendorf tube at a temperature of −80 °C. Upon completion of the study, the samples were conveyed to the Medical Microbiology Laboratory at Istanbul Bezmialem Vakif University, which offered the requisite conditions. The research employed the Enzyme-Linked Immunosorbent Assay technique to evaluate Human Galectin-3 utilising kits from eBioscience in Austria, Europe/International.

## Liver ultrasonography and liver stiffness assessments

All patients had liver ultrasound screening utilising a high-resolution USG device (Philips EPIQ 7) with a 1- to 5-MHz high-resolution convex probe (Philips Health Care, Bothell, WA, USA). A liver ultrasound was conducted following a minimum fasting period of 8 h, utilising B-mode ultrasound in greyscale to evaluate liver dimensions and parenchymal echogenicity. The liver's longitudinal dimensions were assessed at the midclavicular line by measuring the maximum craniocaudal length while the patient was in a supine position. In

typical circumstances, hepatic echogenicity displays a homogeneous echo pattern, which is comparable to or slightly more echogenic than that of the normal renal cortex or spleen (grade 0). Distinct visualisation of the hepatic and portal vein walls exhibiting greater echogenicity than the kidney or spleen was classified as mild steatosis (grade 1 hepatosteatosis). Blurred hepatic and portal vein walls accompanied by increased echogenicity were categorised as moderate steatosis (grade 2 hepatosteatosis). Severe steatosis (grade 3 hepatosteatosis) was identified by posterior attenuation, characterised by the inability to visualise the posterior hepatic segments due to significant shadowing, along with the inability to differentiate the diaphragm. Participants were assessed separately by two seasoned radiologists.

LS measurements were conducted utilising the FibroScan® Mini 430 device (Echosens, France). Subjects were assessed separately by two seasoned professionals in internal medicine. FibroScan was deemed effective only when a minimum of 10 valid readings were acquired, and the interquartile range (IQR) to median ratio of these values was ≤0.3. The LS levels of the individuals were quantified in kilopascals (kPa). In this study, LS more than 8 kPa was established as an indicator of substantial liver fibrosis. Participants were categorised into two groups according to fibrosis levels: below and over 8 kPa.

## Statistical analysis

All analyses were conducted utilising the statistical software program SPSS 24.0 (IBM Corp., Armonk, NY, USA). The Kolmogorov-Smirnov test was employed to evaluate the normality of the distribution of continuous variables. Continuous variables in grouped data were represented as mean ± standard deviation. Categorical variables were represented as numerical values and percentages. The Student's t-test or one-way ANOVA was employed to compare continuous variables having a normal distribution across groups. The Mann-Whitney U test was employed to compare continuous variables lacking normal distribution. The Chi-square ($\chi2$) test was employed to compare categorical variables. The κ coefficient was employed to assess the interobserver and intraobserver variability of ultrasound measurements. A univariate correlation analysis was conducted using the Pearson-Spearman correlation approach to identify the factors related to liver stiffness in participants. A multivariate model was employed for linear regression analysis, incorporating statistically relevant characteristics. Independent variables influencing liver stiffness were identified. A univariate analysis was conducted to independently identify individuals with liver stiffness >8 kPa among those with prediabetes and diabetes mellitus, followed by a multivariate model and multivariate logistic regression analysis, focussing on statistically significant parameters with a $p$-value < 0.05. The threshold for statistical significance was established at $p < 0.05$.

## RESULTS

The study groups were divided into three: NGM (group 1), prediabetes (group 2), and DM (group 3). It was found that the Fib-4 score and LS values were statistically significantly

**Table 1 Demographic, clinical, laboratory findings, galectin-3 levels and liver stiffness of patients with ngm, pre-dm and type-2 dm group.** Values with significant *p* values are indicated in bold.

| Variables | NGM (group 1) (*n* = 60) | Pre-DM (group 2) (*n* = 60) | Patient with type 2 DM (group 3) (*n* = 60) | *p* |
|---|---|---|---|---|
| Age (year) | 58.2 ± 7.11 | 60.3 ± 7.78 | 59.6 ± 8.06 | 0.333 |
| Gender (M/F, *n*) | 31/29 | 32/28 | 36/24 | 0.628 |
| Systolic blood pressure (mmHg) | 116.8 ± 4.68[a,b] | 120.9 ± 7.81[c] | 129.1 ± 13.0 | **<0.001** |
| Diastolic blood pressure (mmHg) | 63.1 ± 4.68[a,b] | 69.5 ± 7.22 | 66.9 ± 6.41 | **<0.001** |
| Body mass index (kg/m$^2$) | 24.5 ± 3.43[a,b] | 30.4 ± 4.28[c] | 28.4 ± 3.58 | **<0.001** |
| Waist circumference, cm | 90.1 ± 3.59[a,b] | 98.7 ± 3.78 | 98.8 ± 3.85 | **<0.001** |
| Basal heart rate (pulse/minute) | 76.0 ± 8.07 | 72.5 ± 8.96 | 75.2 ± 8.49 | 0.117 |
| Fasting plasma glucose, mg/dL | 87.8 ± 8.17[a,b] | 114.9 ± 7.79[c] | 175.1 ± 70.5 | **<0.001** |
| HbA1c, % | 5.51 ± 0.57[a,b] | 6.23 ± 0.33[c] | 8.47 ± 2.23 | **<0.001** |
| White blood cell (10$^3$/μL) | 7.37 ± 1.83 | 7.65 ± 1.31 | 7.12 ± 1.75 | 0.170 |
| Hemoglobin (g/dL) | 13.1 ± 1.72 | 13.3 ± 1.34 | 13.7 ± 1.68 | 0.183 |
| Platelet (10$^3$/μL) | 335.8 ± 50.4[a,b] | 254.8 ± 15.7[c] | 206.6 ± 50.6 | **<0.001** |
| Creatinine (mg/dL) | 0.58 ± 0.13[a,b] | 0.58 ± 0.07 | 0.59 ± 0.08 | **0.016** |
| Sodium (mmol/L) | 138.2 ± 3.34 | 137.0 ± 3.15 | 137.9 ± 2.83 | 0.087 |
| Potassium (mmol/L) | 4.47 ± 0.31 | 4.40 ± 0.42 | 4.57 ± 0.40 | 0.061 |
| Calcium (mg/dL) | 9.22 ± 0.47 | 9.22 ± 0.55 | 9.35 ± 0.47 | 0.324 |
| Aspartate aminotransferase (u/L) | 14.8 ± 1.79[a,b] | 25.5 ± 14.9[c] | 36.1 ± 16.7 | **<0.001** |
| Alanine aminotransferase (u/L) | 27.09 ± 3.02[b] | 27.9 ± 8.97[c] | 35.9 ± 11.0 | **<0.001** |
| Triglycerides, mg/dL | 105.4 ± 44.4[a,b] | 179.3 ± 98.7 | 190.8 ± 101.7 | **<0.001** |
| HDL cholesterol, mg/dL | 52.4 ± 9.75[a,b] | 40.3 ± 8.67 | 39.1 ± 7.70 | **<0.001** |
| LDL cholesterol, mg/dL | 111.6 ± 16.4[b] | 123.3 ± 39.0[c] | 151.3 ± 21.2 | **<0.001** |
| Total cholesterol | 176.4 ± 28.5[a,b] | 195.5 ± 51.5 | 200.4 ± 30.1 | **<0.001** |
| Triglycerides/HDL | 2.09 ± 1.08[a,b] | 4.76 ± 2.98 | 5.23 ± 3.79 | **<0.001** |
| LDL/HDL | 2.20 ± 0.51[a,b] | 3.10 ± 1.12[c] | 4.01 ± 0.98 | **<0.001** |
| Total cholesterol/HDL | 3.47 ± 0.83[a,b] | 5.06 ± 1.78 | 5.31 ± 1.46 | **<0.001** |
| Non-HDL/HDL | 2.47 ± 0.83[a,b] | 4.06 ± 1.78 | 4.31 ± 1.46 | **<0.001** |
| TSH (mIU/L) | 1.70 ± 1.18 | 1.67 ± 1.03 | 2.01 ± 1.28 | 0.243 |
| CRP (mg/L) | 1.21 ± 0.74[b] | 1.53 ± 0.94 | 2.0 ± 0.96 | **0.001** |
| Galectin-3 (ng/mL) | 8.70 ± 1.76[a,b] | 15.3 ± 7.98 | 17.2 ± 7.03 | **<0.001** |
| Fib-4 index | 0.19 ± 0.04[a,b] | 0.49 ± 0.35[c] | 0.71 ± 0.48 | **<0.001** |
| CC liver size, cm | 12.3 ± 1.00[a,b] | 15.2 ± 1.64 | 15.3 ± 1.57 | **<0.001** |
| Liver steatosis grade 1/grade 2–3, n | 0/0[a,b] | 32/9 | 30/16 | **<0.001** |
| Liver stiffness, kPa | 3.18 ± 1.20[a,b] | 5.72 ± 2.53[c] | 7.01 ± 1.68 | **<0.001** |

Notes:
HDL, high density lipoprotein; LDL, low density lipoprotein; CRP, c reaktif protein; TSH, Thyroid stimulating hormone; Fib-4, fibrosis-4; DM, diabetes mellitus; NGM, normal glucose metabolism; CC, Cranio-Caudal.
a = statistical significance between group 1 and group 2.
b = statistical significance between group 1 and group 3.
c = statistical significance between group 2 and group 3.

**Table 2 Evaluation of the study groups according to liver stiffness measurement.** Values with significant *p* values are indicated in bold.

| Variables | <8 kPa (*n* = 132) | ≥8 kPa (*n* = 48) | *p* |
|---|---|---|---|
| Age (year) | 57.8 ± 7.34 | 63.8 ± 6.79 | **<0.001** |
| Gender (M/F, *n*) | 68/64 | 31/17 | 0.116 |
| Systolic blood pressure (mmHg) | 119.7 ± 7.58 | 128.9 ± 13.8 | **0.001** |
| Diastolic blood pressure (mmHg) | 63.1 ± 4.68 | 69.5 ± 7.22 | **0.003** |
| Body mass index (kg/m2) | 27.4 ± 4.74 | 28.9 ± 3.53 | **0.020** |
| Waist circumference, cm | 94.8 ± 5.69 | 98.9 ± 3.88 | **<0.001** |
| Basal heart rate (pulse/minute) | 74.7 ± 8.40 | 74.9 ± 9.43 | 0.926 |
| Fasting plasma glucose, mg/dL | 111.3 ± 35.7 | 166.3 ± 75.2 | **<0.001** |
| HbA1c, % | 6.40 ± 1.45 | 7.73 ± 2.39 | **0.001** |
| White blood cell ($10^3$/μL) | 7.27 ± 1.64 | 7.67 ± 1.66 | 0.150 |
| Hemoglobin (g/dL) | 13.4 ± 1.62 | 13.4 ± 1.54 | 0.731 |
| Platelet ($10^3$/μL) | 286.2 ± 59.9 | 209.4 ± 56.2 | **<0.001** |
| Creatinine (mg/dL) | 0.74 ± 0.20 | 0.85 ± 0.31 | **0.005** |
| Sodium (mmol/L) | 137.7 ± 3.16 | 137.6 ± 3.12 | 0.864 |
| Potassium (mmol/L) | 4.48 ± 0.38 | 4.49 ± 0.39 | 0.805 |
| Calcium (mg/dL) | 9.23 ± 0.49 | 9.34 ± 0.5 | 0.242 |
| Aspartate aminotransferase (u/L) | 18.6 ± 4.29 | 44.3 ± 19.4 | **<0.001** |
| Alanine aminotransferase (u/L) | 29.6 ± 9.12 | 33.4 ± 8.80 | **0.014** |
| Triglycerides, mg/dL | 152.4 ± 95.6 | 175.1 ± 85.4 | 0.149 |
| HDL cholesterol, mg/dL | 46.2 ± 10.4 | 37.7 ± 8.63 | **<0.001** |
| LDL cholesterol, mg/dL | 123.4 ± 30.2 | 143.4 ± 32.1 | **<0.001** |
| Total cholesterol | 189.0 ± 39.9 | 195.6 ± 36.3 | 0.330 |
| Triglycerides/HDL | 3.66 ± 2.89 | 5.04 ± 3.67 | **0.013** |
| LDL/HDL | 2.81 ± 1.04 | 3.94 ± 1.14 | **<0.001** |
| Total cholesterol/HDL | 4.31 ± 1.45 | 5.46 ± 1.82 | **<0.001** |
| Non-HDL/HDL | 3.31 ± 1.45 | 4.46 ± 1.82 | **<0.001** |
| TSH (mIU/L) | 1.70 ± 1.20 | 2.04 ± 1.10 | 0.108 |
| CRP (mg/L) | 1.52 ± 0.90 | 1.75 ± 1.03 | 0.227 |
| Galectin-3 (ng/mL) | 11.2 ± 4.43 | 20.6 ± 8.74 | **<0.001** |
| Fib-4 index | 0.29 ± 0.14 | 0.94 ± 0.50 | **<0.001** |
| CC liver size, cm | 13.9 ± 1.99 | 15.3 ± 1.64 | **<0.001** |
| Liver steatosis grade 1/grade 2–3, n | 40/13 | 22/12 | **<0.001** |
| Pre-DM/DM, n | 47/25 | 13/35 | **<0.001** |

Note:
HDL, high density lipoprotein; LDL, low density lipoprotein; CRP, c reaktif protein; TSH, Thyroid stimulating hormone; Fib-4, fibrosis-4; DM, diabetes mellitus; CC, Cranio-Caudal.

higher in group 3 compared to groups 1 and 2, and in group 2 compared to group 1. Galectin-3 levels, liver size, and the number of patients with grade 2–3 hepatosteatosis were statistically significantly higher in groups 2 and 3 compared to group 1 (Table 1).

The study groups were also divided into two based on LS values: <8 and >8 kPa. It was found that age, BMI, FBG, HbA1c, galectin-3, Fib-4 score, liver size, the number of DM

**Table 3 Multivariate logistic regression analysis for the detection of patients with liver stiffness ≥8 kPa.** Values with significant $p$ values are indicated in bold.

| Variables | Odds ratio | 95% confidence interval | $p$ |
|---|---|---|---|
| Galectin-3 (ng/mL) | 1.235 | 1.129–1.350 | **<0.001** |
| Waist circumference, cm | 1.069 | 0.955–1.197 | 0.244 |
| Triglycerides/HDL | 0.858 | 0.712–1.036 | 0.111 |
| LDL/HDL | 4.249 | 1.946–9.278 | **<0.001** |
| Total cholesterol/HDL | 0.710 | 0.427–1.182 | 0.188 |
| Body mass index (kg/m$^2$) | 0.998 | 0.880–1.132 | 0.976 |

**Note:**
HDL, high density lipoprotein; LDL, low density lipoprotein.

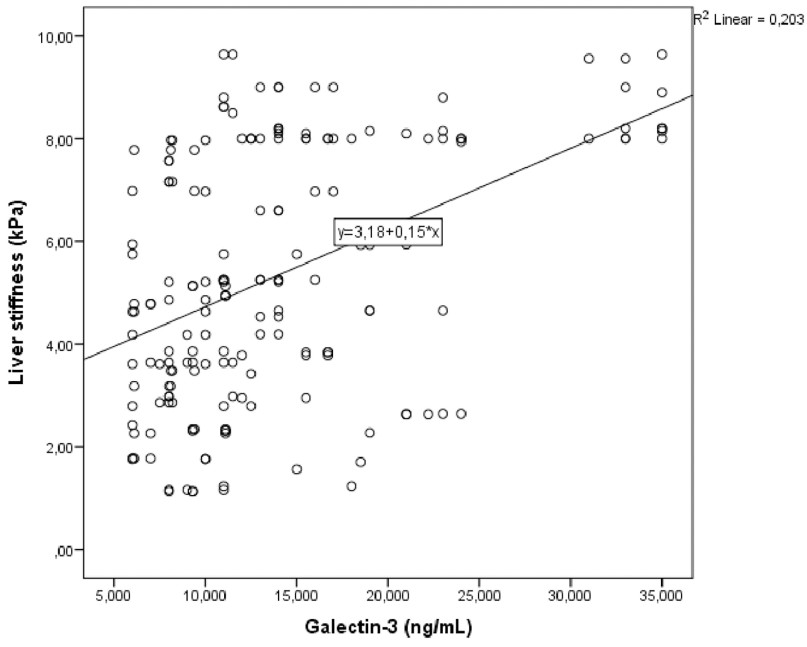

**Figure 1 Scatter plot diagram between liver stiffness and serum galectin-3 level.**

patients and the number of patients with grade 2–3 hepatosteatosis were higher in the group with >8 kPa (Table 2).

A multivariate logistic regression analysis was conducted on the patient groups (groups 2 and 3), using parameters that were found to be statistically significant in the univariate analysis for patients with LS ≥ 8 kPa ($p$ value <0.05). Multivariate logistic regression analysis found that each 1-unit increase in serum galectin 3 level increased the probability of LS ≥ 8 kPa by 23%. It was found that each 1 unit increase in LDL/HDL increased the probability of LS ≥ 8 kPa by 4.2 times (Table 3).

Correlation analysis was performed between LS and other demographic, clinical and laboratory parameters in study group. In univariate analysis, parameters associated with LS were galectin 3, fib-4 score, HbA1c and triglycerides/HDL. Linear regression analysis was

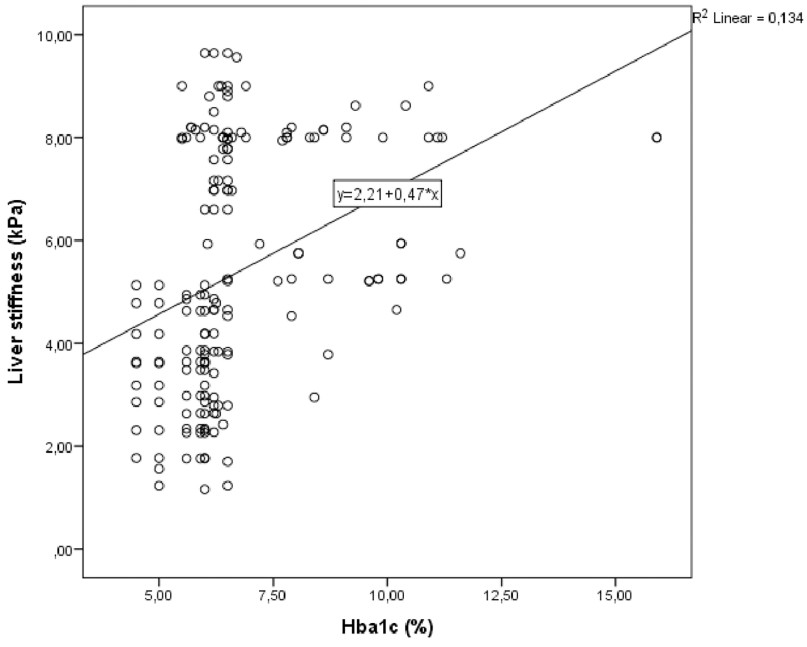

**Figure 2 Scatter plot diagram between liver stiffness and HbA1c level.**

**Table 4 The parameters associated with liver stiffness measurements.** Values with significant $p$ values are indicated in bold.

| | Univariate analysis | | Multivariate analysis | |
|---|---|---|---|---|
| | $p$ | r | $p$ | β |
| Galectin-3 (ng/mL) | <0.001 | 0.426 | 0.003 | 0.063 |
| Fib-4 index | <0.001 | 0.626 | <0.001 | 2.950 |
| HbA1c, % | <0.001 | 0.359 | 0.033 | 0.170 |
| Triglycerides/HDL | <0.001 | 0.306 | 0.019 | 0.105 |

**Note:**
$R^2$ adjusted = 0.463

performed for parameters that were significantly associated with LS in univariate analysis. As a result of this analysis, it was determined that galectin 3 (Fig. 1), HbA1c (Fig. 2), fib-4 index and triglycerides/HDL were related to LS (Table 4).

# DISCUSSION

Our study's main finding is the increase in Fib-4 index, LS and galectin-3 levels in prediabetic patients. Demonstrating that these parameters, which are known to increase in DM patients, are also elevated in prediabetic individuals is significant. Another important finding is that HbA1c and galectin-3 levels were higher in patients with LS ≥ 8 kPa, and both HbA1c and galectin-3 levels were independently associated with LS. Considering the role of galectin-3 in inflammatory and fibrotic processes, the observed relationship

between LS and galectin-3 suggests that inflammatory and fibrotic processes may begin in prediabetes, like DM.

Insulin resistance is a starting point for prediabetes, metabolic syndrome and liver fibrosis. Most prediabetic patients have both insulin resistance and hyperinsulinemia. Prediabetes and MASLD share similar risk factors such as being overweight or obese. These factors contribute to systemic insulin resistance and increased levels of circulating free fatty acids, which are stored in the liver and lead to MASLD. Hepatic fat accumulation increases hepatic insulin resistance, activates inflammatory pathways, increases oxidative stress, and results in hepatic fibrosis. In DM, where insulin resistance is the basis, poor glycemic control is associated with histologically more severe steatohepatitis (*Chen et al., 2017*; *Cooreman, Vonghia & Francque, 2024*). Studies by *Chen et al. (2017)* and *Mahachai et al. (2023)* have reported increased LS in individuals with prediabetes (*Mondal et al., 2018*). In our study, we also evaluated LS using the Fibroscan method. MASLD is associated with an increased risk of cirrhosis, cardiovascular diseases and cancer. MASLD patients remain asymptomatic until severe liver disease develops, making early diagnosis of MASLD crucial to prevent disease progression and associated complications.

Although liver biopsy is the gold standard for diagnosing metabolic dysfunction-associated steatohepatitis (MASH), its invasiveness and potential complications limit its widespread use (*Sumida, Nakajima & Itoh, 2014*). Non-invasive approaches such as Fibroscan have been developed, demonstrating good sensitivity in detecting MASLD and significant fibrosis (*Geethakumari et al., 2022*). In our study, we also found that the Fib-4 index, a simple non-invasive scoring test, was elevated in prediabetic patients. Additionally, patients with LS > 8 kPa had higher Fib-4 index scores, indicating that non-invasive methods like the Fib-4 index and Fibroscan can be used for LS assessment in prediabetic patients.

*Atalar et al. (2019)* found elevated galectin-3 levels in prediabetic and DM patients, suggesting that galectin-3 might be associated with inflammation, metabolic syndrome, and conditions such as insulin resistance and beta-cell dysfunction. Their study indicated that galectin-3 could serve as a marker for the early detection of prediabetes and diabetes. Furthermore, their findings suggested that galectin-3 plays a significant role in the progression of prediabetes to diabetes (*Atalar et al., 2019*). In our study, galectin-3 levels were higher in prediabetic and DM patients compared to NGM patients. While galectin-3 levels were higher in DM patients than in prediabetic patients, the difference was not statistically significant. This result is important as it indicates that the inflammatory process begins in prediabetes.

Evidence from experimental and human studies suggests that galectin-3 plays a critical role in the pathophysiology of MASLD. Galectin-3 has been identified as a biomarker of fibrosis progression in various organs and is generally associated with pro-fibrotic activity due to its ability to regulate the activity of fibroblasts and macrophages in chronically inflamed organs. Additionally, galectin-3 is associated with fibrotic regions in the liver (*Sotoudeheian, 2024*). For MASH, especially in advanced stages of fibrosis and cirrhosis,

new treatments are urgently needed. Galectin-3 inhibition is being explored as a novel antifibrotic therapy for the liver (*Kram, 2023*).

As far as we know from the literature, there are no studies evaluating the relationship between galectin-3 and LS in prediabetic patients. In our study, we found that HbA1c and galectin-3 levels were higher in the group with LS > 8 kPa and that LS was independently associated with HbA1c and galectin-3 levels. Furthermore, we determined that each 1 ng/mL increase in galectin-3 increased the likelihood of LS > 8 kPa by 23%. These findings are important as they demonstrate that as regulation deteriorates in DM, liver fat accumulation and fibrosis develop, leading to increased LS values. Additionally, galectin-3 levels may rise due to increased inflammation and liver fibrosis as DM regulation worsens. In our study, we found that every one unit increase in LDL/HDL increased the probability of LS ≥ 8 kPa by 4.2 times. Lipid metabolism is also closely related to MASLD. Previous studies have shown that higher HDL is an independent protective factor for MASLD, while triglycerides, LDL and total cholesterol are associated with an increased risk of MASLD. Recently, some studies have indicated that the LDL/HDL cholesterol ratio can evaluate LDL and HDL simultaneously and its performance in predicting the risk of cardiovascular metabolic-related diseases is better than that of a single lipoprotein (*Zou et al., 2021*). The LDL/HDL ratio can be used to screen individuals at high risk of MASLD in clinical practice. In our study, we additionally found that LS was independently associated with the triglyceride/HDL ratio. In recent years, it has been stated that the triglyceride/HDL ratio may be a marker of insulin resistance/type 2 diabetes and a predictor for the diagnosis of metabolic syndrome. It has been reported that the prevalence of MASLD is higher in individuals with higher triglyceride/HDL ratio (*Martínez-Montoro et al., 2024*). Triglyceride/HDL ratio may be a useful marker for detecting MASLD in patients with obesity.

MASLD is strongly associated with metabolic abnormalities such as insulin resistance and DM. Previous research has highlighted a bidirectional relationship between MASLD and DM. On one hand, MASLD is a known risk factor for the development of DM and its complications; on the other hand, DM increases the risk of progression to MASH and advanced liver fibrosis. Based on these assumptions, clinical practice guidelines recommend screening for MASLD and advanced liver fibrosis in DM patients using liver fibrosis scores and/or Fibroscan (*Ciardullo & Perseghin, 2022*). The independent association between LS and galectin-3 levels in our study suggests that patients should be closely monitored for MASLD starting from the prediabetes stage and evaluated using these parameters in the early period.

Our study had some limitations. It was a single-center study. New multi-center studies with a larger number of patients are needed. We used the Fibroscan method for LS assessment. New studies could be conducted using magnetic resonance elastography, another sensitive and non-invasive method. We did not classify prediabetic patients based on the duration of their condition, which warrants further follow-up studies. We did not categorize the patients according to medication use, physical activity and diet. There was no use of glucagon-like peptide-1 receptor agonists, which are known to reduce LS, among the patients. However, we did not separate the patients according to the use of sodium

glucose cotransporter two inhibitors and pioglitazone. Similar studies are needed on patients classified according to early diabetes age and medication use or those who did not use any medication.

## CONCLUSION

In conclusion, we found that LS and galectin-3 levels were increased in patients diagnosed with prediabetes and that there was an independent relationship between LS and galectin-3 levels. Considering the increased prevalence of MASLD in prediabetes, we recommend early assessment of LS using non-invasive methods and measurement of galectin-3 levels in these patients.

### Funding
The authors received no funding for this work.

### Competing Interests
The authors declare that they have no competing interests.

### Author Contributions
- Huseyin Ali Ozturk conceived and designed the experiments, performed the experiments, analyzed the data, prepared figures and/or tables, authored or reviewed drafts of the article, and approved the final draft.
- Hilmi Erdem Sumbul conceived and designed the experiments, performed the experiments, analyzed the data, prepared figures and/or tables, authored or reviewed drafts of the article, and approved the final draft.

### Human Ethics
The following information was supplied relating to ethical approvals (*i.e.*, approving body and any reference numbers):

Adana City Training and Research Hospital Ethics Committee approved the study with decision number 3032 dated 21.12.2023.

### Data Availability
The raw measurements are available in the Supplemental File.

### Supplemental Information
Supplemental information for this article can be found online at http://dx.doi.org/10.7717/peerj.19377#supplemental-information.

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
