# Peer review of "Liver stiffness and serum galectin 3 level significantly increases in patients with prediabetes: a fibroscan study"

_PeerJ, doi:10.7717/peerj.19377_

## Round 0.1 · original submission · Major Revisions

· Academic Editor

Major Revisions

Please response to the reviewers point by point.

**Language Note:** The review process has identified that the English language must be improved. PeerJ can provide language editing services - please contact us at [email protected] for pricing (be sure to provide your manuscript number and title). Alternatively, you should make your own arrangements to improve the language quality and provide details in your response letter. – PeerJ Staff

·

Basic reporting

1.The quality of English language needs sprucing up.
2. It is suggested that literature references should also include publications on MASLD with reference to insulin resistance and glycemic control. This would paint a better picture, for the simple reason that insulin resistance forms the core of Type 2 Diabetes mellitus. The authors to please note the same.

Experimental design

1. The authors to please mention, though in brief, the methods/procedures in Clinical Biochemistry that comply with IFCC guidelines
2. A line or two on the Quality control measures adopted in the estimation of the analytes
3. The authors to please compute lipid ratios based on the lipid profile that they have undertaken. This would provide a better picture. The authors may please link the lipid ratios to the fibroscan data and enzymes

Validity of the findings

The validity of the findings would get enhanced, if the authors could please respond to the suggestions/recommendations of the reviewer. The conclusion will also be more robust, if the authors take cognizance of the suggestions given

Additional comments

Insulin resistance is such an important facet in diabetes mellitus as well as in prediabetes. The characteristic indices on insulin resistance would have added greater value to the manuscript

·

Basic reporting

The manuscript addresses a clinically relevant gap in understanding early hepatic changes in prediabetes. To my knowledge, this is the first study evaluating galectin-3 and LS specifically in prediabetes. The study demonstrates that LS and galectin-3 elevations begin in prediabetes, suggesting early hepatic fibro-inflammatory processes.
The manuscript is well written. However, I have two suggestions that should be stated in the manuscript as study limitations:
1. Key confounders (e.g., diet, physical activity, medication use) were not adjusted for, which may influence LS and galectin-3 levels.
2. BMI and metabolic syndrome components (e.g., lipid profiles) were mentioned but not analyzed as covariates in multivariate models.

Experimental design

no comment

Validity of the findings

no comment

---

## Round 0.2 · accepted · Accept

· Academic Editor

Accept

All the comments were well addressed.

·

Basic reporting

The authors addressed all questions. I have no additional comments.

Experimental design

No comment.

Validity of the findings

No comment.

Additional comments

No comment.